# The Induction of Antigen 85B-Specific CD8^+^ T Cells by Recombinant BCG Protects against Mycobacterial Infection in Mice

**DOI:** 10.3390/ijms24020966

**Published:** 2023-01-04

**Authors:** Shihoko Komine-Aizawa, Satoru Mizuno, Akira Kawano, Satoshi Hayakawa, Kazuhiro Matsuo, Mitsuo Honda

**Affiliations:** 1Division of Microbiology, Department of Pathology and Microbiology, Nihon University School of Medicine, 30-1 Oyaguchi-kamicho, Tokyo 173-8610, Japan; 2Japan BCG Laboratory, 3-1-5 Matsuyama, Tokyo 204-0022, Japan

**Keywords:** tuberculosis, BCG, recombinant BCG, prime–boost vaccination, CTL

## Abstract

*Mycobacterium tuberculosis* (Mtb) infection remains a major health problem worldwide. Although the Bacillus Calmette-Guérin (BCG) vaccine is the most widely used vaccination for preventing tuberculosis (TB), its efficacy is limited. We previously developed a new recombinant BCG (rBCG)-based vaccine encoding the Ag85B protein of *M. kansasii* (Mkan85B), termed rBCG-Mkan85B, and its administration is followed by boosting with plasmid DNA expressing the Ag85B gene (DNA-Mkan85B). Previously, we identified MHC-I (H2-Kd)-restricted epitopes that highly cross-react with those of Mtb in BALB/c (H2d) and CB6F1 (H2b/d) mice. We also reported that the rBCG-Mkan85B/DNA-Mkan85B prime–boost vaccination protocol protected CB6F1 mice against *M. kansasii* infection. In this study, to investigate the protective effect of our novel rBCG against Mtb infection, CB6F1 mice were either left unimmunized or immunized with the BCG, rBCG-Mkan85B, or rBCG-Mkan85B/DNA-Mkan85B vaccine for 10 weeks prior to inhalation exposure to the virulent Mtb Erdman strain for another 6 weeks. Compared with the BCG and rBCG-Mkan85B vaccinations, the rBCG-Mkan85B/DNA-Mkan85B prime–boost vaccination protocol significantly reduced the numbers of pulmonary colony-forming units (CFUs). Moreover, the rBCG-Mkan85B/DNA-Mkan85B prime–boost vaccination induced antigen-specific polyfunctional CD4^+^ and CD8^+^ T cells. These results suggest that CD8^+^ T-cell immunity to immunodominant epitopes of Mtb is enhanced by rBCG vector-based immunization. Thus, rBCG vector-based vaccinations may overcome the limited ability of the current BCG vaccine to elicit TB immunity.

## 1. Introduction

*Mycobacterium tuberculosis* (Mtb) continues to cause major health concerns worldwide, and the development of a more effective vaccine is needed. The WHO estimates that one-quarter of the world’s population is infected with Mtb and that approximately 9.9 million people have developed tuberculosis (TB). Several respiratory infections, such as influenza, decreased in frequency during the COVID-19 pandemic, and although the number of new TB infections appeared to decrease in 2020, this figure increased again in 2021. Recently, TB has become an even more serious public health problem worldwide with the emergence of multidrug-resistant (MDR), extensively resistant (XDR), and totally resistant (TDR) Mtb strains.

We previously developed a new recombinant BCG (rBCG) vaccine encoding the *M. kansasii* (rBCG-Mkan85B) antigen 85B (Ag85B) protein. We used plasmid DNA expressing the same *M. kansasii* Ag85B gene (DNA-Mkan85B) as a booster vaccination [1]. The Ag85 complex is composed of three abundantly secreted proteins (Ag85A, Ag85B, Ag85C) that exhibit cell wall mycolyltransferase activity and play an essential role in mycobacterial pathogenesis [2]. The Ag85 proteins are reported to be able to induce a potent cellular immune response against mycobacteria, such as *M. tuberculosis* and *M. leprae* [3,4]. We reported that the rBCG-Mkan85B vaccine expressed Ag85B at levels 9.3-fold higher than those found with the parental BCG vaccine [1]. We also revealed that the rBCG-Mkan85B/DNA-Mkan85B prime–boost vaccination induced antigen-specific polyfunctional CD4^+^ T cells and CD8^+^ T cells [1] and identified two MHC-I (H2-Kd)-restricted epitopes that induced cross-reactive responses to Mtb and other related mycobacteria, such as *M. avium*, *M. kansasii*, and *M. leprae,* in both BALB/c (H2d) mice and CB6F1 (H2b/d) mice [1]. We also confirmed that the rBCG-Mkan85B/DNA-Mkan85B prime–boost vaccination effectively protected CB6F1 mice against intratracheal infection with *M. kansasii* [5]. Therefore, in the present study, to investigate the efficacy of rBCG-Mkan85B/DNA-Mkan85B, we challenged BCG-, rBCG-Mkan85B- and rBCG-Mkan85B/DNA-Mkan85B prime–boost-vaccinated CB6F1 mice with *M. tuberculosis* and evaluated the protective effect and cellular immunity.

## 2. Results and Discussion

CB6F1 mice were either left unimmunized or immunized with BCG-Tokyo, rBCG-Mkan85B, or rBCG-Mkan85B/DNA-Mkan85B for 10 weeks prior to inhalation exposure to the virulent Mtb Erdman strain for another 6 weeks (62.5 ± 33.0 CFUs/mouse) (Figure 1). Naïve CB6F1 mice produced 2.8 × 10^4^ pulmonary CFUs per animal (Figure 2). All three vaccination strategies were effective for reducing the number of pulmonary CFUs in the mice (*p* < 0.01) (Figure 2). The most effective protection induced by prior vaccination was observed in the animals primed with rBCG-Mkan85B and boosted with DNA-Mkan85B (*p* < 0.05–0.01) (Figure 2).

To analyze the immune response induced by the vaccines, we also examined the induction of antigen-specific polyfunctional T cells using an intracellular cytokine staining (ICS) method. Immune cells isolated from the lungs and splenocytes were obtained from each mouse infected with Mtb either unvaccinated or vaccinated with BCG-Tokyo, BCG-Mkan85B, or rBCG-Mkan85B/DNA-Mkan85B. The cells were stimulated with 9-mer Ag85B CD8 epitope peptides (Pep8) [1,5] and the 15-mer Ag85B CD4 epitope peptide (peptide 25) [6] in vitro. T cells producing two (IFN-γ+IL-2, IFN-γ+TNF, or IL-2+TNF) or three (IFN-γ+IL-2+TNF) cytokines were defined as polyfunctional T cells. The rBCG-Mkan85B vaccination induced increases in the levels of peptide 25-specific polyfunctional CD4^+^ T cells in both the lung and spleen (Figure 3A,B). In contrast, Pep8-specific polyfunctional CD8^+^ T cells were strongly induced only after rBCG-Mkan85B/DNA-Mkan85B prime–boost vaccination (Figure 3A,B). 

The Bacillus Calmette-Guérin (BCG) vaccine, which is a live attenuated strain of *M. bovis*, is the only available vaccine for Mtb. Although the BCG vaccine has been used for approximately 100 years, researchers have reached a consensus that the effectiveness of the BCG vaccine in preventing TB infection is limited. Therefore, the development of a novel TB vaccine is urgently needed. The development of various new TB vaccines has been widely pursued, and several different rBCG vaccines and immunization strategies have been explored. Various methods have been attempted to construct new rBCG vaccines, including inserting mycobacteria-derived antigens into the conventional BCG vaccine [1,7,8,9,10] or deleting genes from the BCG strain to increase phagolysosome delivery [11,12] and modifying cyclic dinucleotide metabolism [13]. Some of these candidate rBCG vaccines have advanced to clinical trials. For example, rBCG-expressing listeriolysin O (VPM1002), which disrupts phagosomal membranes, is currently being tested in a phase III clinical trial [12,14]. In addition to attempts to develop new TB vaccines using the rBCG strategy, several viral vector-based vaccines have recently been developed. For example, MVA85A is a recombinant viral vector-based vaccine that expresses the Mtb immunodominant antigen Ag85A and was the first of the new TB vaccines to complete phase IIb clinical trials [15]. In addition to the MVA85A vaccine, various viral vector-based vaccines have been developed [16].

To protect against Mtb, the induction of effective cellular immunity is important. However, although the conventional BCG vaccine can elicit antigen-specific CD4^+^ T cells, the induction of antigen-specific CD8^+^ T cells is poor. Therefore, some novel rBCG vaccines have been designed to induce TB antigen-specific CD8^+^ T cells in addition to TB antigen-specific CD4^+^ T cells [1,17]. We previously reported that rBCG-based vectors have been used in a variety of applications to induce T-cell immunity [18]. We and other researchers reported that rBCGs are more effective in eliciting Mtb-specific immune responses, including polyfunctional antigen-specific CD8^+^ T cells, than the BCG vaccine alone [1,7,19]. In this study, we found that rBCG-Mkan85B vaccination alone could reduce the bacterial burden in the lungs. However, the protective effect was not significantly different from that of the parental BCG vaccine. Although the rBCG-Mkan85B vaccine alone induced a stronger induction of peptide 25-specific polyfunctional CD4^+^ T cells in both the lungs and spleen, this vaccination strategy could not induce polyfunctional CD8^+^ T cells (Figure 3A,B). However, the increased induction of polyfunctional CD8^+^ T cells was observed in rBCG-Mkan85B/DNA-Mkan85B prime–boost-vaccinated mice. Moreover, the rBCG-Mkan85B/DNA-Mkan85B prime–boost vaccination protocol significantly reduced the numbers of pulmonary CFUs. This result suggests that polyfunctional antigen-specific CD8^+^ T cells induced by the rBCG-Mkan85B/DNA-Mkan85B prime–boost vaccination protocol are effective for TB defense. 

The present study has several limitations that need to be considered. Firstly, the long-term effects of the rBCG-Mkan85B/DNA-Mkan85B prime–boost vaccine were not examined in this study. TB is a chronic respiratory infection, and latent TB is an important clinical issue. In the future, the persistence of the effects of the rBCG-Mkan85B/DNA-Mkan85B prime–boost vaccine should be examined. Secondly, we should examine CD4^+^ or CD8^+^ T cells induced by the rBCG-Mkan85B/DNA-Mkan85B prime–boost vaccination in more detail. In the present study, we could not examine the detailed properties of the vaccine-induced CD4^+^ or CD8^+^ T cells, from the viewpoint of central memory and effector memory phenotypes. We would also like to further investigate the T-cell repertoire of the CD4^+^ and CD8^+^ T cells induced by the rBCG-Mkan85B/DNA-Mkan85B prime–boost vaccination protocol. 

In conclusion, we demonstrated that the rBCG-Mkan85B/DNA-Mkan85B prime–boost vaccination protocol protected CB6F1 mice against Mtb infection. The rBCG-Mkan85B/DNA-Mkan85B prime–boost vaccination induced antigen-specific polyfunctional CD8^+^ T cells in addition to polyfunctional CD4^+^ T cells. These results suggest that the prime–boost vaccination protocol with an rBCG expressing Ag85B derived from *M. kansasii* and DNA that also expresses Ag85B derived from *M. kansasii* may effectively increase the population of antigen-specific T cells and thereby induce a more efficient control of Mtb infection.

## 3. Materials and Methods

### 3.1. Animals

Specific pathogen-free female CB6F1 mice, aged between 6 and 8 weeks, were purchased from Japan SLC Inc. (Shizuoka, Japan). All animal studies were conducted in accordance with institutional guidelines approved by the Nihon University Animal Care and Use Committee (AP16M004-1, AP17M057-1, AP19MED029-3, AP19MED056-2), the Institutional Committee for Gene Recombination Experiments (2018MED21), and the Biorisk Management and Control Committee (30-10-4). The institutional animal experimental guidelines comply with the Institute of Laboratory Animal Resources (ILAR) Guide. The mice were allowed free access to sterile water and standard mouse food, and their physiological conditions were assessed every few days. 

### 3.2. BCG and rBCG-Mkan85B Culture

We used the previously prepared rBCG-Mkan85B strain and DNA-Mkan85B plasmid [1,5] in this study. BCG was cultured in Middlebrook 7H9 broth (BD Difco, Franklin Lakes, NJ, USA) supplemented with albumin dextrose complex (ADC) enrichment (BD Difco) and 0.05% Tween 80 at 37 °C. The rBCG-Mkan85B strain was cultured in Middlebrook 7H9 broth (BD Difco) supplemented with ADC enrichment (BD Difco), 0.05% Tween 80, and 100 mg/mL kanamycin at 37 °C. The bacterial culture density was monitored by measuring the absorbances at 470 nm and 600 nm.

### 3.3. Immunization

CB6F1 mice were immunized with either the BCG or rBCG-Mkan85B vaccine at a concentration of 4 × 10^6^ CFUs, or 0.1 mg of bacilli, intradermally (i.d.), and 100 µg of plasmid DNA or control DNA in saline intramuscularly (i.m.) three times (Figure 1). We used six to ten mice per group. The experiments were performed independently six times.

### 3.4. Mtb Infection

The virulent Mtb Erdman strain was grown in Middlebrook 7H9 broth (BD Difco) supplemented with albumin dextrose complex (ADC) enriched (BD Difco) with 0.05% Tween 80 at 37 °C and harvested at the stage of exponential growth stage. The bacilli were passed through a 5.0 µm filter to obtain a single-cell suspension and frozen at −80 °C until use. Thawed aliquots were diluted in PBS containing 0.05% Tween 20 to the desired inoculum concentrations. Two weeks after the final immunization, the mice were exposed to Mtb in an Inhalation Exposure System (model No.4212 (Glas-Col, Terre Haute, IN, USA). Based on a preliminary study estimating appropriate counts of primary tuberculoma in the lung at autopsy, we adjusted the dose to deliver 62.5 ± 33.0 CFUs of bacilli to the lungs of each animal after Mtb challenge infection. The thoracic and abdominal cavities of the animals were then opened aseptically, and the lungs and spleens were removed. Both the lungs and the spleens were homogenized with a gentleMACS dissociator (Miltenyi Biotec, Bergisch Gladbach, North Rhine-Westphalia, Germany). Tenfold serial dilutions of homogenized lung tissues were inoculated onto duplicate 7H10-OADC agar plates (Difco) to determine the bacterial loads. The colonies were counted after incubation for 3 weeks at 37 °C. The Mtb challenge experiments were conducted in the ABSL3 facility of the Research Institute of Tuberculosis, Tokyo, Japan with the approval from its Animal Research Committee.

### 3.5. Polychromatic Flow Cytometric Analysis of Intracellular Cytokine Production

Intracellular cytokine staining was performed as previously reported [1,5]. In brief, immune cells isolated from the lung and spleen were stimulated with a 9-mer Ag85B CD8 epitope peptide (YYQSGLSIV) (Pep8) [1] and a 15-mer Ag85B CD4 epitope peptide (peptide-25: FQDAYNAAGGHNAVF) [6] for 6 h. The cells were then incubated with LIVE/DEAD Fixable Dead Cell Stains (Thermo Fisher scientific, Waltham, MA, USA) to identify dead cells, followed by surface staining with the antibodies APC-conjugated anti-CD3 (145-2C11, BioLegend, San Diego, CA, USA), PerCP-Cy5.5-conjugated anti-CD8 ((53-6.7, BioLegend), and PE-Cy7-conjugated anti-CD4 (GK1.5, BioLegend). The cells were then fixed and permeabilized using BD Cytofix/Cytoperm (BD Biosciences, Franklin Lakes, NJ, USA) and stained for IFN-γ (PE) (XMG1.2, BD Biosciences), IL-2 (APC-Cy7) (JES6-5H4, BD Biosciences), and TNF (Alexa Fluor 488) (MP6-XT22, BioLegend). The gating strategy used to identify cytokine-producing CD4^+^ and CD8^+^ T cells is shown in Figure 3C. Polyfunctional cells were defined as those producing two or more cytokines using Boolean combinations Figure 3D. FACS analysis was performed using a FACSVerse flow cytometer (BD Biosciences) with FlowJo software (version 10) (BD Biosciences). 

### 3.6. Data Analysis and Statistic

All comparisons between the control group and the vaccinated groups and among the vaccinated groups were conducted by one-way ANOVA with the Tukey-Kramer post hoc test using JMP software (SAS Institute, Cary, NC, USA). The data are expressed as the means ± SDs.

## Figures and Tables

**Figure 1 ijms-24-00966-f001:**
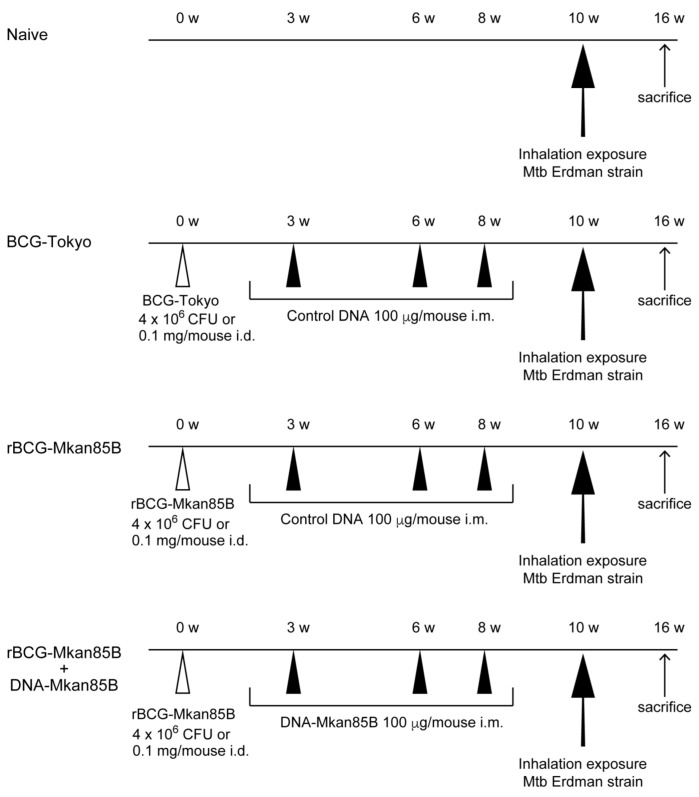
Immunization schedule. CB6F1 mice were immunized with either the BCG vaccine or the rBCG-Mkan85B vaccine at a concentration of 4 × 10^6^ CFUs or 0.1 mg of bacilli i.d., and administered 100 μg of plasmid DNA or control DNA in saline i.m. three times. Six to ten mice were included in each group. The experiments were performed six times independently.

**Figure 2 ijms-24-00966-f002:**
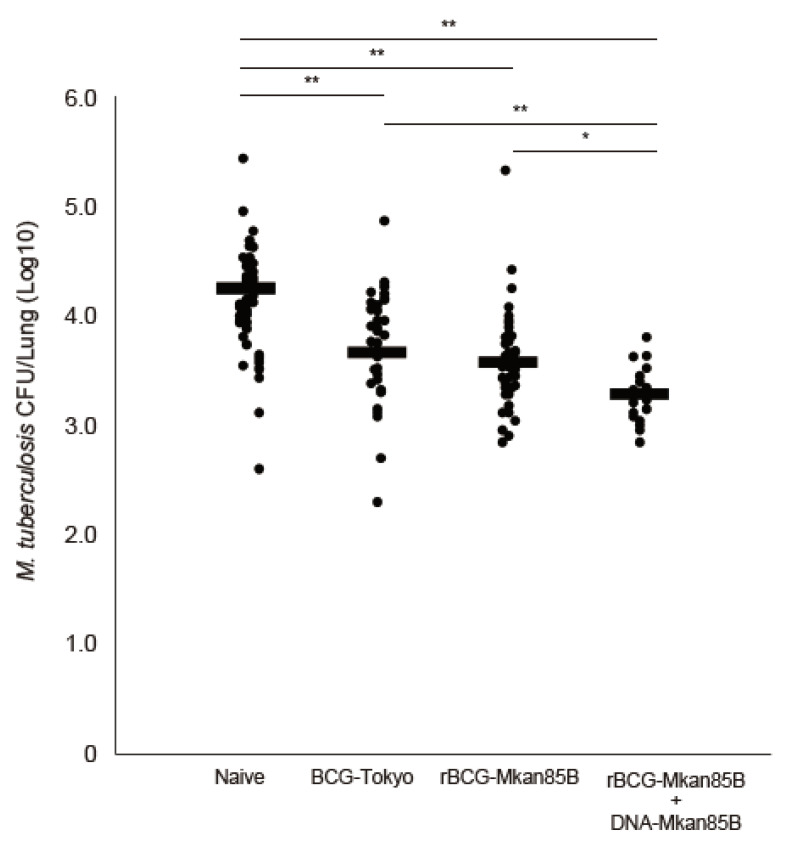
Pulmonary CFUs of unimmunized or immunized with BCG, rBCG-Mkan85B, or rBCG-Mkan85B/DNA-Mkan85B mice 6 weeks after Mtb Erdman challenge. The data represent the results from six independent experiments with six to ten mice per group. The detection limit for bacilli in the tissue homogenate was 15 CFUs. * *p* < 0.05. ** *p* < 0.01.

**Figure 3 ijms-24-00966-f003:**
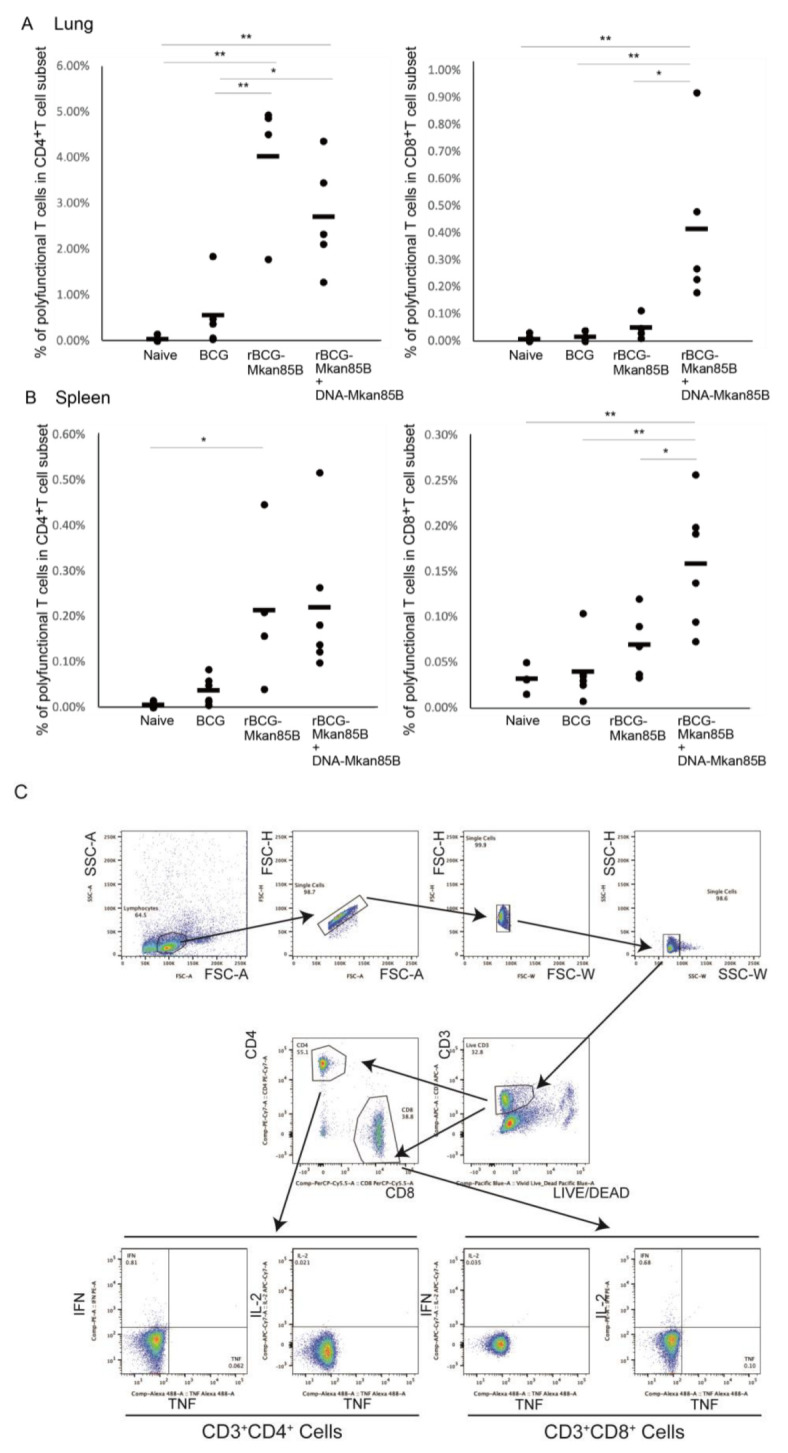
Polyfunctional CD4^+^ and CD8^+^ T cells in (**A**) immune cells isolated from lung (**B**) splenocytes of BCG-, rBCG-Mkan85B-, or rBCG-Mkan85B/DNA-Mkan85B-vaccinated CB6F1 mice. The induction of polyfunctional CD4^+^ T cells by stimulation with peptide 25 (left panel) and the induction of polyfunctional CD8^+^ T cells by stimulation with Pep8 (right panel) are shown. The data represent the results from three independent experiments with two to four mice per group. * *p* < 0.05. ** *p* < 0.01. (**C**) The gating strategy used to identify IFN-g-, IL-2- and TNF-producing CD4^+^ and CD8^+^ T cells among immune cells isolated from lung and splenocytes from a representative mouse is shown. The upper 4 panels show the initial gating of total events, including a singlet cell gate, followed by selection for lymphocytes. Live CD3^+^ T cells were identified by negative LIVE/DEAD staining and CD3 expression. CD4^+^ and CD8^+^ T cells were further identified by CD4 or CD8 expression. Antigen-specific IFN-g-, IL-2- and TNF-producing CD4^+^ and CD8^+^ T cells were gated as shown. The cells producing three, any two and only one cytokine were determined by Boolean combinations. Each cytokine-positive cell was assigned to one of seven possible combinations of these three cytokines (**D**), and the total number of cytokine-producing cells was calculated as the percent of CD4- and CD8-positive cells. The sum of the three- and two-cytokine-producing cells was measured as polyfunctional T cells specific for epitope peptides (**D**).

## Data Availability

The authors declare that the data supporting the findings of this study are available within the paper.

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
