# Peer review of "The Induction of Antigen 85B-Specific CD8^+^ T Cells by Recombinant BCG Protects against Mycobacterial Infection in Mice"

_ijms, 2023, doi:10.3390/ijms24020966_

Round 1

Reviewer 1 Report

I enjoyed reading your research article. It is well thought experiment which also addressed the issue of previous Ag85B based vaccine producing protective effect and the need to examine the central memory and effector memory phenotypes and the T-cell repertoire. I appreciate your foresight to address the existing issue, also to plan for further improvements. I suggest minor changes such as.,

1)      In the result and discussion, second paragraph line seven, states that they used three cytokines to define the polyfunctionality but did not mention the names of the cytokines. Even though it is mentioned in the figure, it is good to have them clearly stated on the result section.

2)      The quality of the figure 3 C must be improved

3)      Polyfunctionality of T cells were chosen by either two or three of the cytokine TNFa, INFg and IL2. It would be nice to show a plot like spice chart to understand which of the cytokine’s are expressed abundantly especially by CD4. In other studies, they showed that high secretion of IFNg, induces CD4 to produce IL-10.

4)       It would be nice to mention the clone details for the antibodies used for the flow cytometry assay.

Good Luck!

Reviewer 2 Report

Overall, the manuscript is well written. While the results obtained from these studies are potentially interesting, several questions arose while reading the manuscript. The following points should be addressed by the authors.

Major –

1. The author should report the lung histopathology score to confirm the protective efficacy against challenge with Mtb in BCG-primed rMkan85B DNA-boosted mice.

2. The author should look at the memory T cells phenotype to support their findings.

3. It appears that the experiment in Fig 3 has only been once with 4 mice per group. It is important to conclude data from at least 2-3 independent experiments. Please repeat.

4. Both the introduction and the discussion lack information about vector-based vaccination already been reported. I think the author should improve both of them by providing literature already available.

Minor

1. The gating strategy in Figure 3C is fuzzy and the resolution needs to be improved.

2. Line 120- Remove "to" in the sentence “The present study has several limitations that need to consideration”

3. Line 111- 113 Sentence not clear “Although the ……..insufficient”. Please rephrase.

Reviewer 3 Report

Comments for the authors

Point 1: In this research article, the authors investigate the protective effect of their novel rBCG against Mtb infection. BCG, rBCG-Mkan85B,or rBCG-Mkan85B/DNA-Mkan85B vaccine were used for immunization of the CB6F1 mice, 10 weeks prior to inhalation exposure to the virulent Mtb Erdman strain for another 6 weeks. Compared with the BCG and rBCG-Mkan85B vaccinations, the rBCG-Mkan85B/DNA-Mkan85B prime–boost vaccination significantly reduced pulmonary colony-forming units (CFUs), induced antigen-specific polyfunctional CD4+ and CD8+ T cells. The manuscript is well written and the short story is easy to follow; the experimental design and data analysis are robust.

Figure 1 immunization schedul is clear, However could you specfiy the  inclusion of  CB6F1 mice immunization for this experimental deisgn, would be better to draw detaild experimental workflow combine it with figure 1.

Point 2: Figure 2 is pulmonary CFUs after Mtb infection, would be nice to compare bacterial load of short and long term Mtb infection after immunization. Is there any statistcal significance among BCG and  rBCG-Mkan85B  CFUs.

Point 3: In figure 3 Polyfunctional CD4+ and CD8+ T cells responses were compared,  rBCG- Mkan85B/DNA-Mkan85B prime–boost vaccination induced antigen-specific polyfunctional CD8+ T cells in addition to polyfunctional CD4+ T cells. It would be interesting to show whether there is an impact in the long-term, including in the establishment of a memory response followed by an Mtb challenge to determine the protective role of the recall response in upcoming experiments. Futhermore if cells were stimulated with PMA or and anti-CD28 both ex vivo and in vivo would be nice  to add their details if possible, any specific CD4+ T cells peptied/tetramer were used in this study . Overall, I could not really fault the experiments or the interpretation. Future experiments in memory response would be informative.

Round 2

Reviewer 2 Report

I would like to confirm that I am satisfied with the revised manuscript. The author and team have addressed all the comments and suggestions and have made the necessary changes required. Therefore, the manuscript should be accepted for publication.